# Fumed SiO$_2$-H$_2$SO$_4$-PVA Gel Electrolyte CO Electrochemical Gas Sensor

**Yuhang Zhang** [1], **Dongliang Cheng** [1], **Zicheng Wu** [1], **Feihu Li** [1,2], **Fang Fang** [1,2,*] and **Zili Zhan** [1,2,*]

1   School of Chemical Engineering, Zhengzhou University, No.100 of Science Road, Zhengzhou 450001, China; zhangyuhang950110@163.com (Y.Z.); C931715884@163.com (D.C.); 18848962133@163.com (Z.W.); lifeihu@zzu.edu.cn (F.L.)

2   Engineering Research Center of Advanced Functional Material Manufacturing of Ministry of Education, No. 100 of Science Road, Zhengzhou 450001, China

*   Correspondence: fangf@zzu.edu.cn (F.F.); zhanzili@zzu.edu.cn (Z.Z.)

**Abstract:** The conventional CO electrochemical gas sensor uses aqueous H$_2$SO$_4$ solution as electrolyte, with inevitable problems, such as the drying and leakage of electrolyte. Thus, research on new alternative electrolytes is an attractive field in electrochemical gas sensors. In this paper, the application of a new fumed SiO$_2$ gel electrolyte was studied in electrochemical gas sensors. The effects of fumed SiO$_2$ and H$_2$SO$_4$ contents on the performance of the CO gas sensor were investigated. The results showed that the optimized composition of the SiO$_2$ gel electrolyte was 4.8% SiO$_2$, 38% H$_2$SO$_4$, and 0.005% polyvinyl alcohol (PVA). Compared with aqueous H$_2$SO$_4$, the gel electrolyte had better water retention ability. The signal current of the sensor was proportional to the CO concentration. The sensitivity to CO was 78.6 nA/ppm, and the response and recovery times were 31 and 38 s, respectively. The detection limit was 2 ppm. The linear range was from 2 to 500 ppm. The gel electrolyte CO sensor possesses equivalent performance to that with aqueous electrolyte.

**Keywords:** fumed SiO$_2$; gel electrolyte; electrochemical sensor; CO gas sensor

---

## 1. Introduction

Toxic, flammable, explosive and polluting gases in the environment pose a huge threat to human life and property safety. Thus, the online monitoring of gases is necessary. Gas sensors are effective for realizing fast and real-time monitoring of these dangerous gases. At present, the most widely used gas sensors on the market are semiconductor metal oxide, catalytic combustion, optical, and electrochemical sensors [1,2]. Korotcenkov comprehensively compared the advantages and disadvantages of various gas sensors [1]. Mahajan et al. pointed out the disadvantages of different types of gas sensors [2]. The few major disadvantages are as follows. Semiconductor metal oxide sensors operate at high temperature. Catalytic combustion sensors have low selectivity and risk of catalyst poisoning. Optical sensors have difficulty in miniaturization and high cost. Among these sensors, the electrochemical sensor, which has the advantages of room temperature operation, good linearity, and high measurement accuracy, is one of the main sensors used for accurate quantitative monitoring [3].

The electrochemical gas sensor has the above advantages, but usually uses aqueous H$_2$SO$_4$ solution as electrolyte. The conventional electrochemical gas sensor has problems such as water loss or absorption [4], resulting in short lifetime and poor long-term stability. Therefore, searching for alternative electrolytes has become an important aspect in electrochemical gas sensors. The study of alternative electrolytes has been focused on ionic liquids [5], polymers [6] and inorganic solid electrolytes [7]. Considering their low saturated vapor pressure and good stability, ionic liquids have become one of the hotspots of gas sensor studies in recent years and are expected to solve the problems

of current aqueous electrolytes. However, the diffusion of reactants in ionic liquids is slow due to high viscosity of ionic liquids, leading to long response time [8]. Nafion is the most widely used polymer electrolyte in gas sensors because of its high proton conductivity, high water diffusivity and good gas permeability [4,9]. Nafion has also been successfully applied to commercialized CO gas sensors, which show good performance to the CO gas [9–11]. The ionic conductivity of Nafion results from the mobility of the hydrated $H^+$ that moves through the polymer film by passing from one $H_2SO_4$ group to another [3]. The sensor needs a reservoir to store water, resulting in the sensor having a large volume [6].

Inorganic gel electrolytes especially those with $SiO_2$ as gelling agent have many advantages over organic colloids, including physical rigidity and high wear resistance, low expansibility in water and organic solutions, chemical inertia, and high thermal stability [12]. However, gel electrolytes have high viscosity and a short gelling time, leading to long electrolyte injection time, and may also lead to uneven distribution of electrolyte. The high viscosity of the colloidal electrolyte may also reduce the diffusion coefficient of gas and ions in the electrolyte and reduce the sensitivity of the sensor. Gel electrolyte containing $SiO_2$ is prepared using the sol–gel method by using colloidal $SiO_2$ or fumed $SiO_2$ as gelling agent. The gelling agents do not participate in the electrochemical reactions within the gas sensor, and their main function is to form a three-dimensional network structure for the entrapment of the aqueous $H_2SO_4$ solution. The gel electrolyte containing $SiO_2$ is widely used in lead–acid batteries [13–15], which possess long lifetime, high reliability under depth of discharge cycles, minimal or even no leakage of acid electrolyte, and minimal corrosion [15].

There is an interaction strong force between water and $SiO_2$ in the gel electrolyte [14], and the gel electrolyte has a high viscosity, so it is expected to have a good water retention ability. The "semi-solid" gel electrolyte containing $SiO_2$ brings hope for solving the problems of dryness and the leakage of conventional aqueous electrolyte. At present, few studies have reported the application of $SiO_2$ gel electrolyte in electrochemical gas sensors for the quantitative analysis of monomethyl hydrazine [16], carbon monoxide [17], hydrogen peroxide [18] and gas chromatographic detectors [7]. The gel electrolytes in these sensors are prepared through the hydrolysis of ethyl orthosilicate to form colloid $SiO_2$. The method has problems such as long gelling time and poor stability and reliability. Gel electrolytes prepared with fumed $SiO_2$ can overcome the above shortcomings [14,19]. Nevertheless, fumed $SiO_2$ is often used as gelling agent in the $SiO_2$ gel electrolyte lead–acid battery, and literature regarding fumed $SiO_2$ gel electrolyte used in gas sensors is not available.

Carbon monoxide is a toxic gas produced during incomplete combustion. CO poisoning is a major public health problem; CO may be responsible for more than one half of all fatal poisonings that are reported worldwide each year [20]. The current Occupational Safety and Health Administration (OSHA) permissible exposure limit (PEL) for carbon monoxide is 50 parts per million (ppm) parts of air (55 milligrams per cubic meter ($mg \cdot m^{-3}$)) as an 8 h time-weighted average (TWA) concentration. CO is impossible to detect by a person because it is colorless, odorless. Therefore, it is very necessary to develop carbon monoxide gas sensor to detect CO concentration in the environment.

In this paper, gel electrolytes were prepared through the sol–gel method by using fumed $SiO_2$ as the gelling agent and aqueous $H_2SO_4$ solution as the liquid phase. Pt black was used as an electrocatalyst in the working (WE), counter (CE), and reference (RE) electrodes. A three-electrode amperometric electrochemical gas sensor with constant potential was fabricated. The effect of the composition of the gel electrolyte in the performance of the CO sensor was investigated, and the sensing performance of the sensor was extensively evaluated. The purpose of this work is to demonstrate the good water retention ability of fumed $SiO_2$ gel electrolyte and to provide an example of the application of fumed $SiO_2$ gel electrolyte in electrochemical gas sensor.

## 2. Experimental

### 2.1. Preparation and Characterization of the Pt Black

99.95% $H_2PtCl_6 \cdot xH_2O$ (trace metals basis) solution and 98% $NaBH_4$ (analytically pure)were purchased from Shanghai Aladdin Biochemical Technology Co. Ltd., Shanghai, China. The Pt black was prepared using the $NaBH_4$ reduction method, and the specific preparation process can be seen in our previous report [21]. 20 mL 10% $H_2PtCl_6$ solution was added into a 200 mL beaker, then 150 mL deionized water was added. The solution was frozen to 0 °C in a refrigerator. 5 g $NaBH_4$ was added into a 500 mL beaker, then the above frozen $H_2PtCl_6$ solution was slowly added into the 500 mL beaker. The reaction produced black precipitate and a lot of bubbles. The precipitation was washed five times with deionized water, filtered, and dried at 50 °C to obtain platinum black catalyst. The morphology of the Pt black catalyst and the screen-printed electrodes were observed using transmission (FEI TalosF200S, Thermo Fisher Scientific, Waltham, MA, USA) and scanning (Auriga-bu, Oberkochen, Germany) electron microscope. The electrode materials of the sensor were characterized using an X-ray diffractometer (D8 Advance, Bruker, Hamburg, Germany).

### 2.2. Preparation of the Gel Electrolyte

The fumed $SiO_2$ (model Hydrophilic-200) with a specific surface area of 200 $m^2 \cdot g^{-1}$ and a particle size of 7–40 nm was purchased from Shanghai Tengzhun Biotechnology Co. Ltd., Shanghai, China. $H_2SO_4$ (98%, analytically pure) and polyvinyl alcohol (PVA chemically pure) were purchased from Shanghai Aladdin Biochemical Technology Co. Ltd., Shanghai, China.

Dilute $H_2SO_4$ with a mass concentration of 60% was prepared and placed in the refrigerator to cool down for later use. The 4.8% $SiO_2$-38%-$H_2SO_4$-0.005% PVA sol electrolyte preparation process was as follows: 35.67 mL deionized water and 5.0420 g $SiO_2$ were added to a 250 mL glass beaker. The mixture was stirred at 2000 r/min for 3 min. Then, 42.39 mL 60% $H_2SO_4$ solution and 1.00 mL 0.05% PVA solution were added and stirred at 600 r/min for 1 h to obtain the sol electrolyte. The PVA solution (0.05%) was prepared in water baths maintained at 99 °C and cooled before use. $SiO_2$ gel solutions with mass content of 9%–25% were prepared on site and used immediately. A certain amount of the as-prepared $H_2SO_4$, $SiO_2$ gel, and PVA solutions were mixed under stirring to prepare gel electrolytes with different content of $SiO_2$ and $H_2SO_4$.

### 2.3. Fabrication and Measurement of the Sensor

The CO electrochemical sensor was a three-electrode structure in the experiment. Pt black was printed on polytetrafluoroethylene film by screen-printing technology to obtain the WE, CE, and RE [21]. The electrodes were prepared with the help of Zhengzhou Winsen Electronic Technology Co. Ltd., Zhengzhou, China. The preparation process of electrodes included screen printing, drying, ethanol washing, deionized water washing, drying and so on. According to the assembly process, the CE, glass mat, RE, glass mat, and WE were loaded into the sensor holder in sequence, which was purchased from Zhengzhou Winsen Electronic Technology Co. Ltd., Zhengzhou, China. The top cover was sealed by ultrasonic bonding. The gel electrolyte was injected into the shell through the injection hole. The injection hole was sealed by ultrasonic bonding. The sensor was stored at room temperature for 24 h until the electrolyte formed a stable gel. Finally, the performance of the sensor was measured after 24 h of electrical aging, as shown in Figure 1.

Standard CO gas samples were purchased from Henan Yuanzheng Science and Technology Development Co. Ltd., China. The sensing performance was measured using a gas sensor test system (EC–CALS00, EC-Sense GmbH, Ebenhausen, Germany). The sensor test system consisted of a gas mixing control unit, a signal acquisition unit, computer control and a storage unit, as shown in Figure 2. The gas mixture control unit included standard CO gas, clean air and a digital flow meter. The required concentration of the CO gas was obtained by controlling the standard CO gas and clean air flow through a digital flow meter. The signal acquisition unit comprised a signal collector and

a sensor adapter connected thereto, and the sensor was inserted into the adapter. When testing the sensor, the required CO concentration and humidity are set by the software on the computer, and the information is received by the digital flow meter and used to automatically adjust CO and air flow. The mixed gas enters the sensor adapter, and the data acquisition system automatically collects and stores sensor signals.

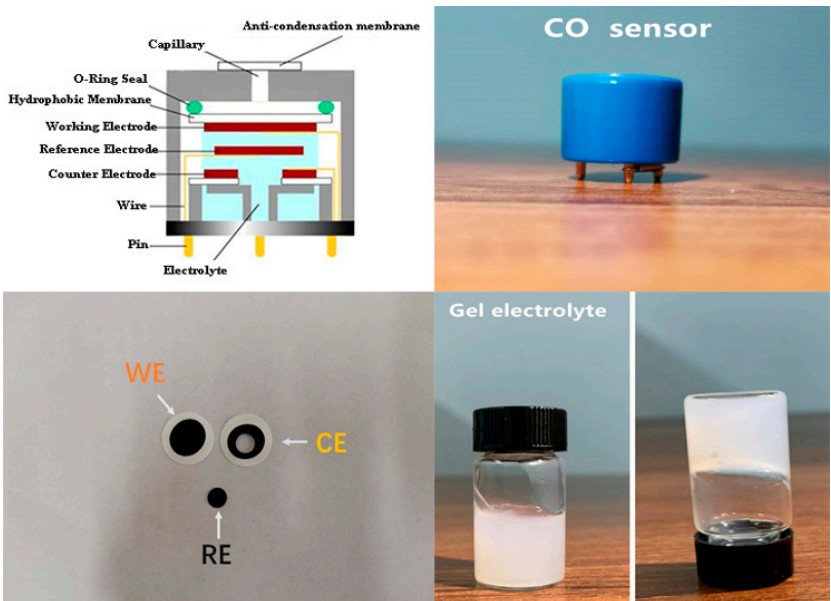

**Figure 1.** The schematic diagram of the sensor structure with the photos of electrodes, sensor and gel.

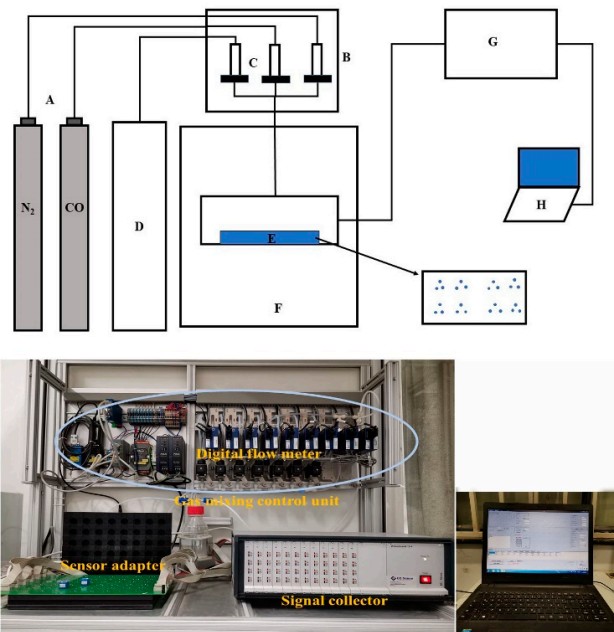

**Figure 2.** The schematic diagram of the sensor test system with the photos of gas mixing control unit, signal acquisition unit, computer control and storage unit. A: Gas cylinder, B: Gas mixing control unit, C: Digital flow meter, D: Clean air, E: Sensor adapter, F: Climate Chamber, G: Signal acquisition unit, H: Computer control and storage unit.

Through software control, the system can complete the sensor performance tests, i.e., gas concentration, temperature and humidity control; sensor signal acquisition; data storage and analysis.

## 3. Results and Discussion

### 3.1. Characterization of Electrode Materials

The TEM image of the Pt black is shown in Figure 3a. The prepared Pt black particles were spherical and uniform. The particles had a size distribution of 5–11 nm, as shown in Figure 3c. The average diameter was 8 nm calculated from TEM image. The crystal lattice was calculated to be 0.225 nm, which is assigned to a (111) plane according to the standard reference from the Joint Committee on Powder Diffraction Standards (PDF01-087-0640). Figure 3b shows the SEM image of the screen-printed WE. The nanosized Pt black particles were dispersed on the surface of the rod-shaped hydrophobic polytetrafluoroethylene. The hydrophilic Pt black and the hydrophobic polytetrafluoroethylene formed a gas diffusion electrode favorable for the gas liquid solid three-phase contact.

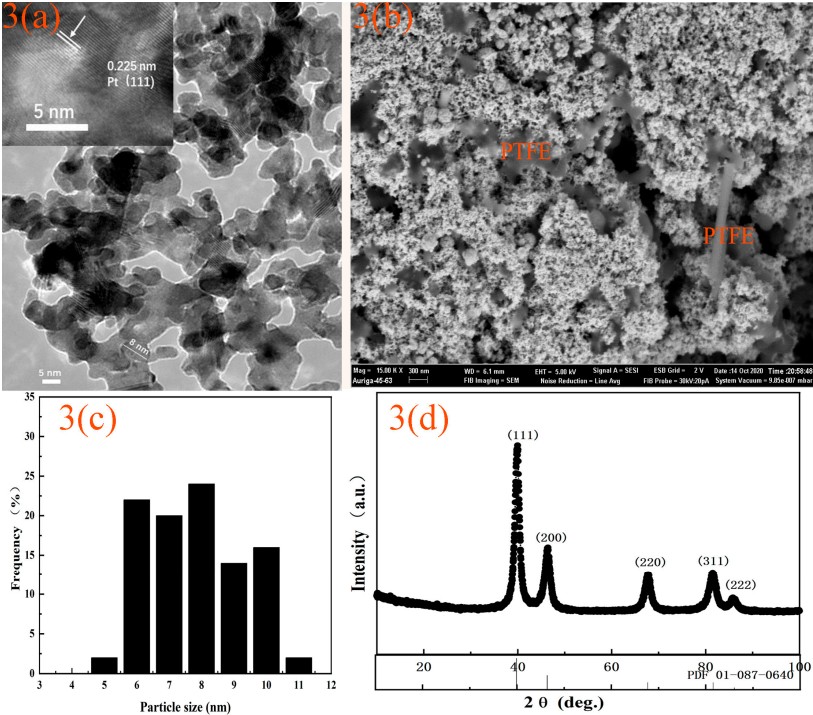

**Figure 3.** Characterization of Platinum black and electrode. (**a**) TEM and HRTEM image of Pt black, (**b**) SEM of electrode, (**c**) Size distribution of Pt black particles, (**d**) XRD spectra of the Pt black.

Figure 3d shows the XRD patterns of the as-prepared Pt black. The XRD patterns revealed that Pt black particles agreed well with the JCPDS reference (PDF01-087-0640), in which the diffraction peaks at $2\theta$ = 39.89°, 41.71°, 67.71°, 39.14°, 81.57°, and 86.04° were assigned to the (111), (200), (220), (311), and (222) diffraction planes. These results indicated that the Pt black prepared had a cubic structure. The particle size of the Pt black calculated using the Scherer formula from the strongest diffraction peak was 7.7 nm, which was consistent with the results obtained using TEM.

### 3.2. Gel Preparation

Gel properties, including stability, strength, and gelling time, are important for forming a gel electrolyte suitable for gas sensors. The electrolyte of a commercial CO electrochemical sensor is composed of aqueous $H_2SO_4$ solution. Thus, the effects of $SiO_2$ and $H_2SO_4$ contents on the strength and gelling time of gel electrolyte should be investigated. Many studies prove that $SiO_2$ influences the properties of gel electrolytes. Tantichanakul et al. showed that electrolytes containing 3.0%–4.0% $SiO_2$ took more than 5 h to form a gel, whereas those containing 5.0% and 6.0% $SiO_2$ started to form a gel after 3 and 2 h, respectively [13]. Most researchers believe that electrolyte with 4.0%–6.0% $SiO_2$ has

improved performance [13,14]. Gençten et al. showed that gel structures with 6.0%–8.0% $SiO_2$ are good [19]. The range of the $SiO_2$ content in this study was 4.0%–8.0%, to ensure the formation of a stable gel. According to reports in the literature, the $H_2SO_4$ concentration range in the gel electrolyte is 30%–42% [13–15]. In this experiment, the $H_2SO_4$ concentration range was between 30% and 40%.

Experimental results showed that the stable gel can be formed at PVA of 0.005%, $SiO_2$ of 4.0%–8.0% and $H_2SO_4$ of 30%–40%. The content of fumed $SiO_2$ is a key factor affecting the gel properties. With increasing $SiO_2$ content, the gelling time shortened, and the viscosity increased rapidly. Figure 4 shows the influence of the $SiO_2$ content on gelling time at 38% $H_2SO_4$ and 0.005% PVA. As shown in Figure 4, when the $SiO_2$ content increased from 4.0% to 5.6%, the gelling time decreased rapidly from 360 to 80 min. Further increasing the $SiO_2$ content resulted in slowly decreased gelling time and rapidly increased viscosity. When the $SiO_2$ content was 8%, the gelling time decreased to 35 min.

When the fumed $SiO_2$ is dispersed in aqueous media, the isolated silanols may form hydrogen bonds with both the isolated silanols on other silica particles (weak hydration forces) and water molecules (strong hydration forces). When fumed $SiO_2$ is used in $H_2SO_4$ solution, most of its isolated surface silanols link to form weak hydrogen bonds with each other. This gives a three-dimensional network gel structure, entrapping the $H_2SO_4$ solution [14]. With the increase of $SiO_2$ content, the number of hydrogen bonds increases, resulting in short gelling time and high viscosity.

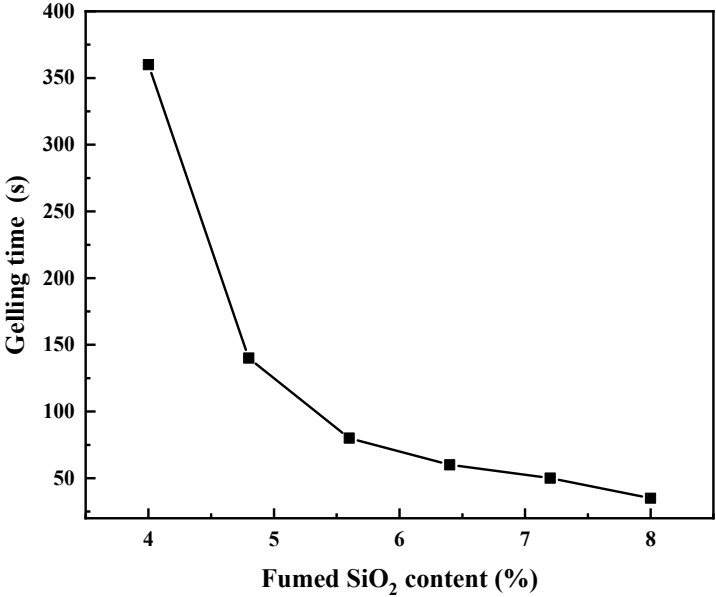

**Figure 4.** Effect of fumed $SiO_2$ content on the gelling time.

EIS of the sensors integrated with the gel and the aqueous $H_2SO_4$ solution electrolyte are compared in Figure 5. Obviously, the resistance of electrolyte of the gel-based sensor reaches 38 Ω, much larger than that of the aqueous $H_2SO_4$-based sensor (3.5 Ω). The major reason is presumably the lower conductivity of the gel. On the other hand, the resistance of charge transference of the gel-based sensor is also larger, reflecting the sluggish electrochemical kinetics in the gel-based electrolyte.

Thermogravimetric analysis (TGA) was performed in air with a heating rate of 10 °C·min$^{-1}$ on Shimadzu DTG-60 instruments, Shimadzu, Japan. Figure 6 shows the typical TGA curves for aqueous $H_2SO_4$ solution and $SiO_2$ gel electrolytes. The TGA curves of aqueous and gel electrolytes the showed apparent weight loss from 40 °C to 170 °C. The weight loss was due to the loss of water in the electrolyte. The weight loss of aqueous $H_2SO_4$ solution is obviously greater than that of colloidal electrolyte. The water loss of aqueous solution was 1.52 times that of gel at 100 °C. The weight loss of water in the gel electrolyte was reduced by 34% at 100 °C. Better water retention of gel electrolyte is beneficial to prolonging the life of the sensor under dry conditions.

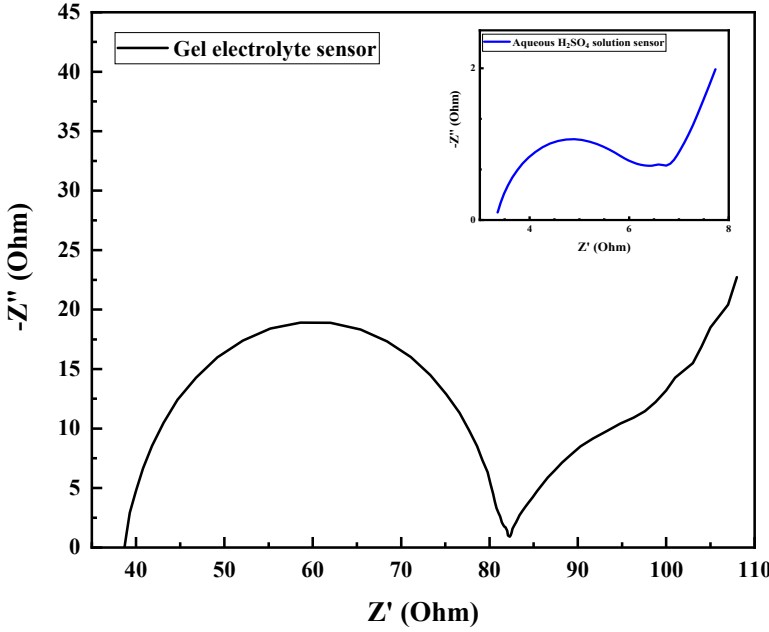

**Figure 5.** Electrochemical impedance spectra (EIS) of the SiO$_2$ gel electrolyte and the aqueous H$_2$SO$_4$ solution sensors.

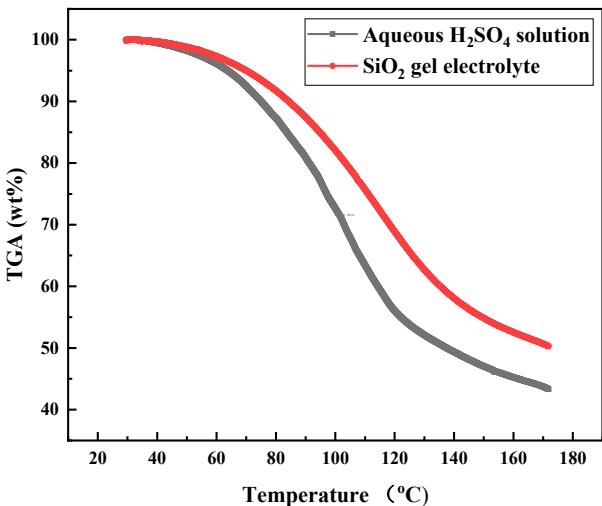

**Figure 6.** TGA curves of the SiO$_2$ gel and aqueous H$_2$SO$_4$ solution electrolytes.

### 3.3. Effect of Fumed SiO$_2$ Content on the Response of CO Sensor

The three-electrode electrochemical gas sensor adopted the structure of WE, CE, and RE, in which no current passing between RE and WE were observed, and RE was used as a standard to stabilize the voltage of WE. At WE potential of 0 V (vs. Pt RE), the CO sensor has the best performance [9]. The commercial electrochemical gas sensor usually uses 0 V bias voltage [22]. In this paper, the amperometric measurements were performed at 0 V vs. Pt RE. It is known from [9–11] that electrode reactions are as follows:

$$\text{WE}: \ 2CO + 2H_2O = 2CO_2 + 4H^+ + 4e^-.$$

$$\text{CE}: \ O_2 + 4H^+ + 4e^- = 2H_2O.$$

$$\text{Total Reaction}: \ 2CO + O_2 = 2CO_2.$$

The content of $SiO_2$ greatly influenced the performance of the gel electrolyte, thereby influencing the performance of the electrochemical gas sensor. Therefore, CO gas sensors with 4.0%, 4.8%, 5.6%, 6.4% and 7.2% fumed $SiO_2$ were prepared, and the PVA and $H_2SO_4$ in the gel electrolyte were fixed at 0.005% and 38%, respectively. The influence of the $SiO_2$ content on the response current $I$ of the CO sensor was investigated Figure 7. With increasing $SiO_2$ content, the response of the sensor to 200 ppm CO first increased and then decreased. When the $SiO_2$ content increased from 4.0% to 5.6%, $I$ increased from 8.4 μA to 19.8 μA. When the $SiO_2$ content was further increased to 6.4%, $I$ decreased to 11.9 μA. At $SiO_2$ content of 7.2%, $I$ decreased slightly. For clarity, the response curve of the sensor with 7.2% $SiO_2$ content was not given in Figure 7. When the $SiO_2$ content reached 8.0%, the gelling time decreased to 35 min, and the viscosity was too high to complete glue injection. The decreased response may be due to the increased $SiO_2$ content, which sharply increased the electrolyte viscosity and significantly shortened the gelling time. This phenomenon resulted in difficult liquid injection and the even distribution of electrolyte in the sensor.

The sensor containing the gel electrolyte with 5.6% $SiO_2$ had the highest sensitivity but had a gelling time of only 80 min. At the same time, electrolyte with high $SiO_2$ content resulted in high viscosity of the electrolyte and long injection time, which brought difficulties to the mass production of the sensors. The sensor with 4.8% $SiO_2$ had high sensitivity to CO, and used as electrolyte the subsequent tests.

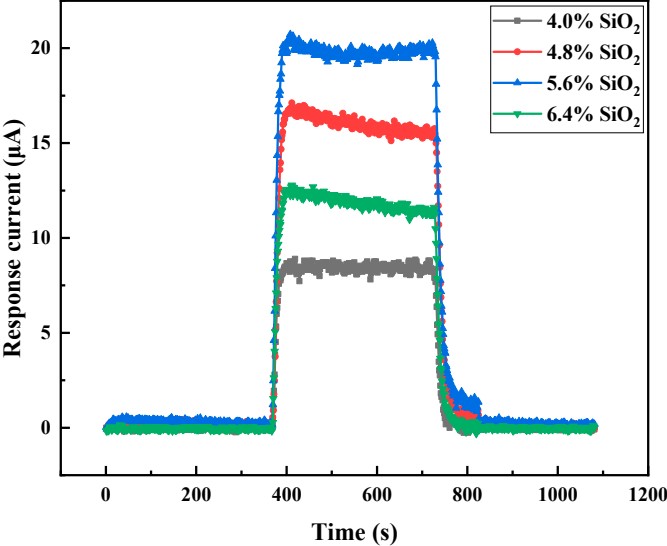

**Figure 7.** Effect of $SiO_2$ content on the response of the gel electrolyte electrochemical CO gas sensor. Test conditions: 25 °C, 50% RH, 200 ppm CO, and 1000 sccm CO.

### 3.4. Effect of $H_2SO_4$ Concentration on the Response of the CO Sensor

The $H_2SO_4$ concentration is another important factor affecting the properties of the gel electrolyte and the gas-sensing performance of sensor. Thus, the effect of $H_2SO_4$ concentration on the CO sensing properties was investigated. The PVA and the fumed $SiO_2$ in the gel electrolyte were fixed at 0.005% and 4.8%, respectively. Gel electrolyte electrochemical gas sensors with $H_2SO_4$ concentration of 32%, 35% and 38% were prepared. The test results are shown in Figure 8.

Figure 8 shows that increasing the $H_2SO_4$ concentration resulted in significantly increased response to 200 ppm CO. When the $H_2SO_4$ concentration increased from 32% to 35% and from 35% to 38%, the response increased by 4.6 and 2.8 μA, respectively. At $H_2SO_4$ concentrations greater than 38%, the viscosity of the solution increased rapidly, and the gelling time shortened, which was not conducive to the liquid injection operation. Thus, the $H_2SO_4$ concentration was not further increased in this experiment and set to 38%.

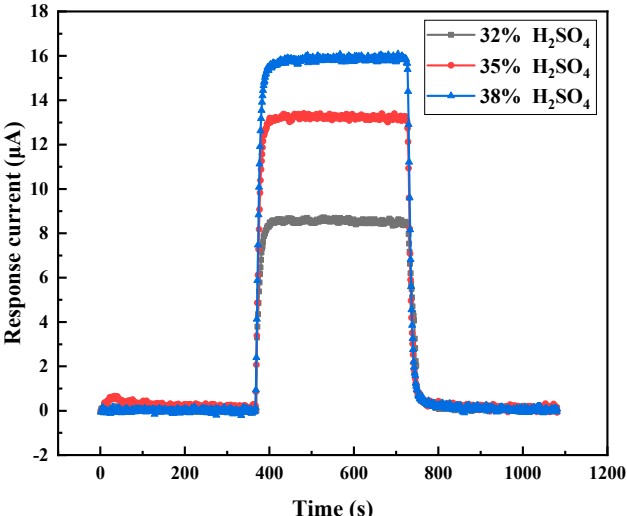

**Figure 8.** Effect of $H_2SO_4$ concentration on the response of the gel electrolyte electrochemical CO gas sensor. Test conditions: 25 °C, 50%RH, 200 ppm CO, and 1000 sccm CO.

### 3.5. Performance of the CO Sensor Based on the $SiO_2$ Gel Electrolyte

Based on the above discussion, gel electrolytes with 4.8% $SiO_2$, 38% $H_2SO_4$, and 0.005% PVA were chosen, and the aqueous electrolyte was 38% $H_2SO_4$ solution. The test parameters were set as follows: 25 °C temperature, 50% relative humidity (RH), and 1000 sccm CO gas flow rate.

The responses of the two sensors to 200 ppm CO were tested to compare the performances of the aqueous $H_2SO_4$ solution and the gel electrolyte sensors. The transient response curve is shown in Figure 9. The response curves of the two sensors had a similar trend, and the response/recovery curves of the two sensors overlapped almost completely. After stabilization, the aqueous $H_2SO_4$ and the gel electrolyte sensor had *I* values of 17.57 µA and 15.71 µA, respectively, and sensitivity values of 87.9 and 78.6 nA/ppm, respectively. The sensitivity of the aqueous $H_2SO_4$ electrolyte sensor was slightly higher than that of the gel electrolyte sensor. The transient response of the gel electrolyte sensor was similar to that of the conventional aqueous $H_2SO_4$ solution electrolyte sensor.

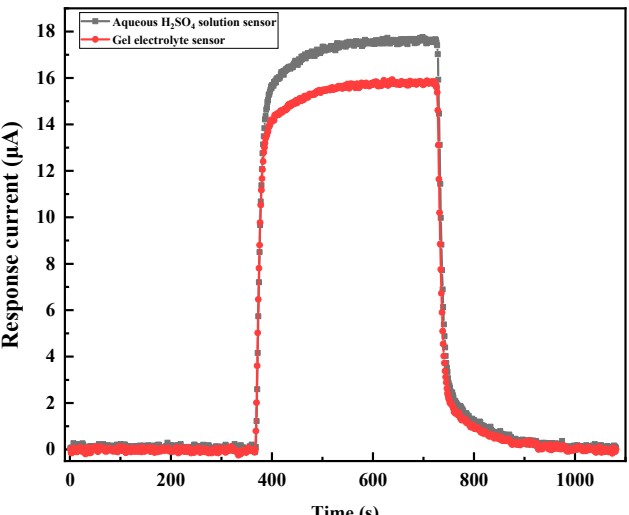

**Figure 9.** Transient response curves of the sensors to 200 ppm CO.

The transient response curves of gel electrolyte sensors towards different concentrations of CO are shown in Figure 10. The baseline of the sensor was stable in air. When 200 ppm CO gas was

injected, the electrochemical reaction occurred, and *I* increased rapidly, reached the maximum value, and entered a stable platform area. After the adoption of air replacement CO, the concentration of CO decreased rapidly, and the *I* signal dropped rapidly. The sensitivity was 78.6 nA/ppm, and the response and the recovery times ($T_{90}$) were 31 and 38 s, respectively. Therefore, the gel electrolyte electrochemical CO gas sensor had excellent response/recovery characteristics.

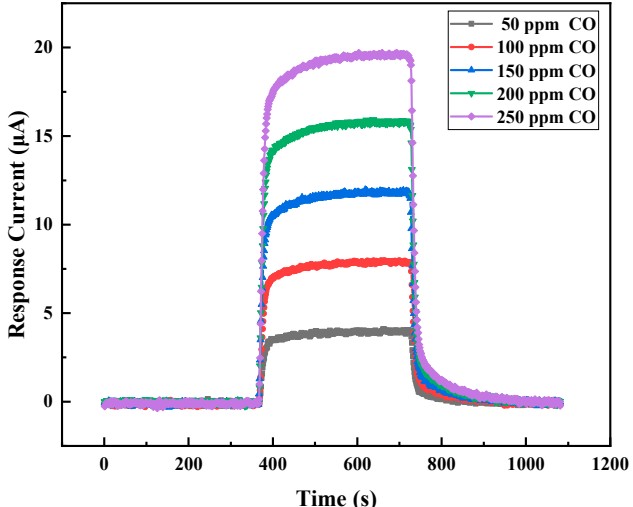

**Figure 10.** Transient response curves of the sensor to different concentrations of CO.

Figure 11 shows the relationship between the *I* of the sensor and the concentration of CO. The signal current of the sensor was proportional to the CO concentration, which was an important advantage of the electrochemical gas sensor and ensured the quantitative detection of CO. The electrode reaction rate was fast because of the high catalytic activity of the nanosized Pt black. The diffusion is the rate-determining step [22,23]. In a steady-state diffusion, the CO diffusion flux can be described by Fick's first Law:

$$J_{co} = -D_{co}\left(\frac{\mathrm{d}C_{co}}{\mathrm{d}x}\right) \tag{1}$$

where $J_{co}$ is the diffusion flux of the CO gas (mol·[cm$^{-2}$·s$^{-1}$]); $D_{co}$ is the diffusion coefficient of the CO gas in the gel electrolyte (cm$^2$·s$^{-1}$), and $\frac{\mathrm{d}C_{co}}{\mathrm{d}x}$ is the concentration gradient of CO (mol·cm$^{-4}$). The negative sign indicates that the direction of diffusion is opposite to that of the concentration increase.

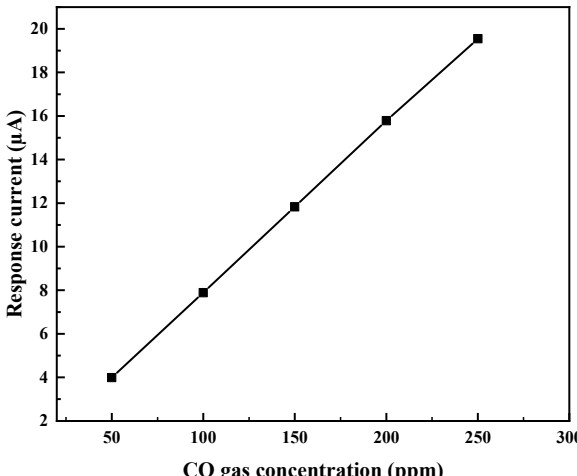

**Figure 11.** Relationship between the response of sensor and the CO gas concentration.

The thickness of the liquid layer on the catalyst surface was constant due to the semi-wetting state of the catalyst surface. The reactant was a flowing gas in the gas phase, The CO concentration in the gas phase could be considered constant. The mass transfer process of the electrochemical gas diffusion electrode was a typical steady-state diffusion. According to Faraday's law, the current density is proportional to the diffusion flux of the CO gas:

$$i = F(J_{co}) = -nFD_{co}\frac{C_{co}^0 - C_{co}^S}{l} \tag{2}$$

where $i$ represents current density (A·cm$^{-2}$), $n$ is the number of electrons exchanged during the electrode process, $F$ is the Faraday constant (C·mol$^{-1}$); $C_{co}^0$ and $C_{co}^S$ are the bulk concentration of CO in the liquid and the surface concentration of the electrode, respectively (mol·cm$^{-3}$), and $l$ is the thickness of the diffusion layer (the thickness of the liquid layer on the surface of the catalyst, cm).

The concentration of CO in the gas phase ($C_{co}^*$) is proportional to the concentration of the bulk CO in the liquid phase ($C_{co}^0$).

$$C_{co}^* = kC_{co}^0 \tag{3}$$

where $k$ is the proportional coefficient, and $A$ is the electrode area. Combining Equations (2) and (3), $I$ is:

$$I = -nFAD_{co}\frac{kC_{co}^* - C_{co}^S}{l} \tag{4}$$

In the rate-determining diffusion process, the electron transfer step can be considered to be a quasi-reversible process, and the surface reaction can be considered to be in an equilibrium state, which conforms to the Nernst equation. $C_{co}^S$ is a constant value at 0 V. Therefore, a linear relationship between the $I$ and CO concentration was observed. As shown in Figure 11, $I$ was directly proportional to the CO gas concentration ($C_{co}^*$), which agreed with the finding that the reactant diffusion was the rate-determining step. Moreover, the linear extension line in Figure 11 intersected with the coordinate (0, 0). From the Formula (4), the CO concentration ($C_{co}^S$) on the electrode surface tended to be zero, indicating the high catalytic activity of the Pt black.

The ambient temperature influenced the performance of the CO gas sensor, whereas the ambient humidity had almost no effect on the sensitivity of the sensor. Therefore, the influences of ambient temperature (−15 °C to 50 °C) on the sensors of conventional aqueous H$_2$SO$_4$ and SiO$_2$ gel electrolytes were compared. The gas-sensing measurement for 200 ppm CO was carried out in 50% RH ambient humidity. The relative sensitivity ($S_r$) is defined as:

$$S_r = \frac{S}{S_0} \times 100\% \tag{5}$$

where $S_0$ represents the sensitivity at 25 °C, and $S$ is the sensitivity at a certain temperature. $S_r$ is based on the sensitivity at 25 °C, and the $S_r$ at other temperatures is obtained. The $S_r$ is 100% at 25 °C.

As shown in Figure 12, the two curves had the same trend, and the sensitivity increased with increasing temperature. These findings were due to the electrode reaction process being controlled by CO diffusion, and the diffusion rate ($D_{co}$) became faster when the temperature increased. The result showed that the $S_r$ increased with increasing ambient temperature. At the same time, Figure 12 shows no significant difference in the effect of ambient temperature in the performance of the two sensors. Therefore, the gel electrolyte sensor had the same resistance to environmental impact as the conventional aqueous H$_2$SO$_4$ electrolyte sensor.

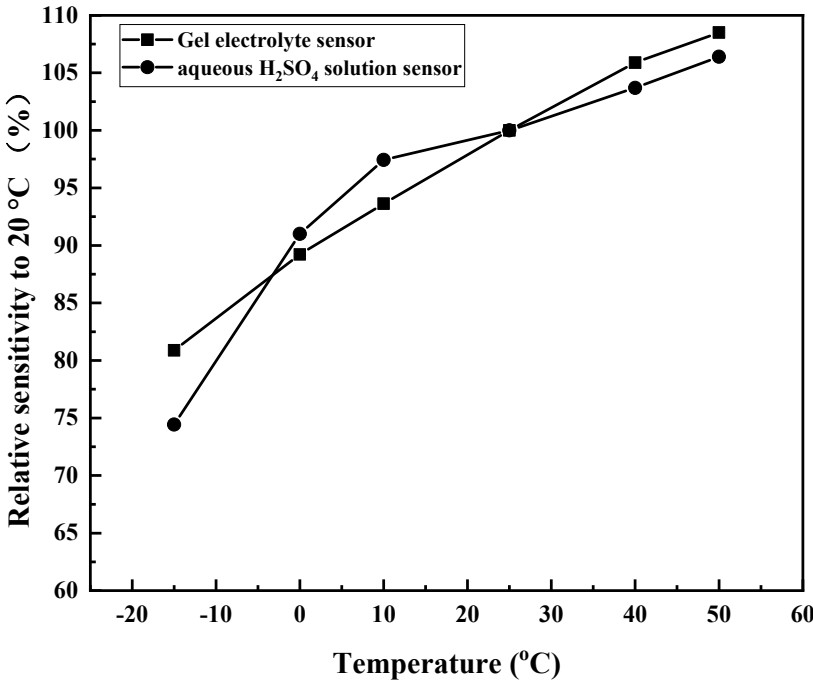

**Figure 12.** Influence of ambient temperature on the relative sensitivity of the two sensors.

## 4. Conclusions

The feasibility of applying fumed $SiO_2$ gel electrolytes in electrochemical gas sensors was studied. A three-electrode amperometric CO gas sensor was fabricated using Pt black as catalyst and fumed $SiO_2$ gel as electrolyte. First, the reasonable composition range of the electrolyte was determined in accordance with the physical properties of the gel and the requirements of the CO electrochemical sensor. Then, according to the performance of the sensor, the composition of the gel electrolyte influenced the *I* of the sensor. The sensors with electrolyte containing 4.8% $SiO_2$ had the maximum sensitivity. With increasing $H_2SO_4$ concentration, the *I* of the sensor increased. However, further increasing the $H_2SO_4$ concentration resulted in rapidly increased solution viscosity, thereby causing difficulties for electrolyte injection. Finally, the optimum composition of the gel electrolyte was 4.8% $SiO_2$, 38% $H_2SO_4$ and 0.005% PVA.

Thermogravimetric analysis showed that the weight loss of water in the gel electrolyte was 34% lower than that of sulfuric acid solution at 100 °C. Good water retention of the gel electrolyte is beneficial to prolonging the life of the sensor under dry conditions. However, gel electrolyte sensors have higher resistance of charge transference than that of aqueous $H_2SO_4$ solution electrolyte sensors.

The new electrolyte may bring new advantages to the electrochemical sensor. Thus, further studying the other properties of the gel electrolyte electrochemical sensor, especially the sensor performance under harsh environment and long-term stability, should be conducted.

**Author Contributions:** Y.Z. conceptualization, methodology, software, formal analysis, data curation, writing—original draft; D.C.; Z.W.; and F.L. software, formal analysis, validation; F.F.; Z.Z. writing—review and editing, supervision. All authors have read and agreed to the published version of the manuscript.

**Funding:** This work was supported by the National Natural Science Foundation of China (22008223), Research program on planar electrochemical gas sensor (20180712, Zhengzhou Winsen Electronic Technology Co. Ltd., Zhengzhou, Henan 450001, China).

**Acknowledgments:** This work was financially supported by Zhengzhou Winsen Electronics Co. Ltd., Zhengzhou, Henan 450001, China.

**Conflicts of Interest:** There are no conflict to declare.

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
