# Peer review of "Fumed SiO2-H2SO4-PVA Gel Electrolyte CO Electrochemical Gas Sensor"

_chemosensors, doi:10.3390/chemosensors8040109_

Round 1

Reviewer 1 Report

The authors studied the alternative electrolyte for CO electrochemical sensor using SiO2 gel. The experimental data and their analysis is reasonable and well organized, so, this manuscript can be published as current form after correcting issues as in below;

  1. at the abstract use the subscription for several chemicals.
  2. please add the device structure with measurement scheme.
  3. Fig.1, Fig. 2, and Fig. 3 can be merged as Fig. 1a, 1b, and 1c for readability
  4. Comments about the faradaic charge transfer resistance. if possible measure and compare the Nyquist type plot between aqueous and gel cases.

Author Response

Dear reviewer 1:

We would like to thank constructive comments and suggestions on our manuscript. Those comments are all valuable and very helpful for revising and improving our paper, as well as the important to our researches. We are enclosing below all the detailed answers point by point. Thank you very much for sharing your opinion and advice.

Point 1: at the abstract use the subscription for several chemicals.

Response 1: Thank you for reminding us to carefully check the subscript. We have checked the subscript in my manuscript and found no error. We speculate that the errors may be caused by different versions of Microsoft Office.

Point 2: please add the device structure with measurement scheme.

Response 2: Figure 2 shows the schematic diagram of the sensor test system with the photos of gas mixing control unit, signal acquisition unit, computer control and storage unit (line 150-152). The measurement scheme was provided in page 2 (line 139-147).

Point 3: Fig.1, Fig. 2, and Fig. 3 can be merged as Fig. 1a, 1b, and 1c for readability.

Response 3: Fig.1, Fig. 2, and Fig. 3 were merged as Fig. 3(a), 3(b), and 3(d) (line 159-161). A new size distribution histogram (Fig. 3c) was inserted in Fig. 3. (line159).

Point 4: Comments about the faradaic charge transfer resistance. if possible measure and compare the Nyquist type plot between aqueous and gel cases.

Response 4: EIS of the sensors integrated with the gel and the aqueous H2SO4 electrolyte are compared in Figure 5 (line 209-211).  The resistance of charge transference of the gel-based sensor is much larger that of the aqueous H2SO4-based sensor (line 212-217).

Reviewer 2 Report

Manuscript title: SiO2 gel electrolyte CO electrochemical gas sensor

This manuscript deals with the use of fumed SiO2 gel as host for H2SO4 electrolyte of commercially designed electrochemical CO sensor. Starting already from the abstract and all throughout the manuscript, there is consistent lack of information on the materials and experimental procedure and methods. Moreover, the manuscript lacks of scientific and technical merits. The sensor that is produced by using the commercially available fumed SiO2 gel electrolyte shows no improvement on comparative tests against the commercial sensor. Despite this fact, the authors grade this fumed SiO2 gel electrolyte containing sensor as having “excellent CO-sensing performance”. In the introduction, the authors indicate a series of disadvantages that the commercial aqueous electrolytes encounter, the essential ones of these are the water loss and absorption and drying and leakage of H2SO4. These results in poor long-term stability and short life-time. The so-called in this work developed sensor is not tested for these properties. There are no long-term stability tests, no signal reproducibility tests after long-term storage or under harsher environments. Thus, no excellence in its properties can be noted. Furthermore, there is no scientific evidence given if and how the fumed SiO2 reacts or interacts with the H2SO4 electrolyte. What other significant role does the fumed SiO2 play in order to prevent H2SO4 dryness or leakage. Also, no experiment has been carried out to demonstrate the superiority of the fumed silica gel against other types of silica gel.   

The editorial deficits of the manuscript are:

Abstract: lacks the essential idea that the manuscript is based on, even, the use of fumed silica is not mentioned

Introduction: The advantages as well as disadvantages of silica gel must have been clearly addressed. Also, these of fumed silica. Why it is not being used commonly?

Experimental:

Provider companies are addresses correctly or missing: There is no SEM equipment called MERLINCO but Merlin Compact from Zeiss. Also, D8 Advance is manufactured by Bruker which is not given at all. Shanghai is in China and this should be written. The provider of the sensor test system is not given. EC-CALS00 seems to be a type of the test instrument but not the provider. The cities of all these providers should be named.

Compositional information: “A certain amount of the as-prepared H2SO4, SiO2 gel, and PVA solutions were mixed”:  This sentence is not appropriate. Even though the experimental findings will be applied, this should have been mentioned.

Page 3:

Experimental: It is not clear if the authors have manufactured a sensor or if they have used a sensor purchased from a Chinese company or if they simply used a standard three-electrode measurement system?  Is a sensor design regenerated from ref 20 or the company Zhengzhou Weisheng manufactures the bought sensor?

Figures 1 and 2: TEM and SEM pictures are not revealing any detail. A higher magnification SEM picture is necessary for more detail and also the size of particles must be marked at TEM. No compositional or elemental hints are given. Pt black is of lose particles and can phase conductivity problems in real systems.

Page 4:

Is it significant that Pt black has cubic structure? On the other hand, more important information is missing. No XRD data from fumed SiO2 or from the used gel SiO2. What types of Silica how influences the gel electrolyte properties.

Ref 20 is on the Battery electrolytes.  

Figure 4: Effect of which silica? please indicate on the graph and also in the figure caption

How are the reactions at WE, CE and total determined? It should be mentioned that it is assumed that the reactions occur and this knowledge is from the references 9 and 12. I appears that the authors themselves found these out.

Figure 5, 6, 7, 8, 10: The types of variants are consistently neglected in the figures as well as in the figure captions; SiO2 concentration, H2SO4, CO, sensor type and sensing test conditions (dry/wet/operating temperature, etc.), which two sensors, Pls indicate in details in graph and in the figure captions.

Page 7:

It is not clear which electrochemical reaction occurs when CO injected, with what it reacts? and how this develops please indicate in the figure. There is only description of it.

Page 10:

There is no improvement against the commercial electrochemical CO-sensor. The manuscript fails to demonstrate the avoidance of long-term disadvantages of aqueous H2SO4 electrolytes that is provided with the applied Fumed SiO2 gel electrolyte

Conclusion: very weak

May bring new advantages: what are these? And how and why it is provided? please indicate clearly.

Author Response

Dear reviewer 2:

We would like to thank the thorough and constructive comments and suggestions on our manuscript. Those comments are all valuable and very helpful for revising and improving our paper, as well as the important to our researches. We are enclosing below all the detailed answers point by point. The revised parts are in red letters in our manuscript. Thank you very much for sharing your opinion and advice.

This manuscript deals with the use of fumed SiO2 gel as host for H2SO4 electrolyte of commercially designed electrochemical CO sensor. Starting already from the abstract and all throughout the manuscript, there is consistent lack of information on the materials and experimental procedure and methods. Moreover, the manuscript lacks of scientific and technical merits. The sensor that is produced by using the commercially available fumed SiO2 gel electrolyte shows no improvement on comparative tests against the commercial sensor. Despite this fact, the authors grade this fumed SiO2 gel electrolyte containing sensor as having “excellent CO-sensing performance”. In the introduction, the authors indicate a series of disadvantages that the commercial aqueous electrolytes encounter, the essential ones of these are the water loss and absorption and drying and leakage of H2SO4. These results in poor long-term stability and short life-time. The so-called in this work developed sensor is not tested for these properties. There are no long-term stability tests, no signal reproducibility tests after long-term storage or under harsher environments. Thus, no excellence in its properties can be noted. Furthermore, there is no scientific evidence given if and how the fumed SiO2 reacts or interacts with the H2SO4 electrolyte. What other significant role does the fumed SiO2 play in order to prevent H2SO4 dryness or leakage. Also, no experiment has been carried out to demonstrate the superiority of the fumed silica gel against other types of silica gel.

Response: First of all, I would like to thank the reviewers for their careful review of the manuscript. Please allow us to explain the above your comments.

  1. As the reviewer pointed out, many experimental processes, especially catalyst preparation and electrode fabrication, are too simple. The preparation of catalyst and electrode was introduced in detail in the master's thesis of my group students (Title: CO Electrochemical Gas Sensors Based on Water-based electrodes,

(https://kns.cnki.net/KCMS/detail/detail.aspx?dbname=CMFD2012&filename=1012353871.nh)   The original cited reference [20] is the dissertation, but the student failed to find the DOI number of the dissertation during the submission process, and replaced the reference 20 with the current semiconductor gas sensor article. It was our mistake, for which I am very sorry. The following modifications were made in the manuscript:

The reference 20 was corrected to be the master's thesis [21] (line 435-437).

The detail information of chemicals and instruments were provided in revised manuscript (line 102-111).

Detail information about Pt catalyst and gel preparation was provided (line 94-101), and the fabrication protocol of the sensor was introduced in the revised manuscript (line 124-130).

The schematic diagram of sensor structure (Figure 1, line 134-136) and the test instrument (Figure 2, line 150-154) were given, and the physical photos were attached.

  1. The initial motivation of using gel electrolyte is to improve the water retention of electrolyte and provide support for small planar electrochemical gas sensor research. The research of gel electrolyte gas sensor has just started. We compared the performance of gel electrolyte and H2SO4 electrolyte sensor under common environmental conditions. Sensor performance under extreme harsh environmental conditions is an important aspect of the research. At present, we have only done the following work:

We compared the thermogravimetric curves of SiO2 gel solution and H2SO4 aqueous solution, and found that the water loss rate of silica gel is significantly lower than that of aqueous H2SO4 solution (Figure 6, line 219-229).

  1. Fumed SiO2 is an exceptionally pure form of silicon dioxide, made by reacting silicon tetrachloride in an oxy-hydrogen flame. The process generates particles in the size range from 7 to 50 nm. When the fumed SiO2 is dispersed in aqueous media, the isolated silanols may form hydrogen bonds with both the isolated silanols on other silica particles (weak hydration forces) and water molecules (strong hydration forces). When fumed silica is used in aqueous H2SO4 solution, most of its isolated surface silanols link to form weak hydrogen bonds with each other. This gives a three-dimensional network gel structure, entrapping the H2SO4 solution [14] (line 201-206).

There is an interaction force between water and SiO2 in the colloidal electrolyte [14], and the gel electrolyte has a high viscosity, so it is expected to have a good water retention ability (line 65-66).

The editorial deficits of the manuscript are:

Point 1: Abstract: lacks the essential idea that the manuscript is based on, even, the use of fumed silica is not mentioned. 

Response 1: Our initial motivation was to design a planar electrochemical gas sensor. The volume of electrolyte pool needs to be reduced. Therefore, the electrolyte with good water retention is needed. The interaction between water molecules and SiO2 in the colloids and the high viscosity of the colloids are expected to delay the dehydration and water absorption rate and prolong the life of the sensors. However, our research is just beginning, and there is no experimental data under harsh conditions. So, there is no mention of this idea in the abstract.

In the abstract, we revised “SiO2” to “fumed SiO2”. It is emphasized that gel electrolyte has good water retention ability (line 18-19).

Point 2: Introduction: The advantages as well as disadvantages of silica gel must have been clearly addressed. Also, these of fumed silica. Why it is not being used commonly?

Response 2: The advantages of colloidal electrolyte have been given in the manuscript (line 53-55). The disadvantages of gel electrolyte were added in the manuscript (line 55-58).

Fumed SiO2 is widely used as filler for strength reinforcement [14], Fumed SiO2 is also used in valve-regulated lead-acid batteries [13-15]. As for the reason that SiO2 is not used in electrochemical gas sensor, we speculate that aqueous H2SO4 electrolyte sensor can meet the requirements under normal environment. Due to the disadvantages of SiO2 gel electrolyte (for example, high viscosity). SiO2 gel electrolyte is more suitable for applications in harsh environments, and the demand of sensors used in harsh conditions is small. Therefore, there are few studies on gel electrolyte sensors.

Point 3: Experimental:

Provider companies are addresses correctly or missing: There is no SEM equipment called MERLINCO but Merlin Compact from Zeiss. Also, D8 Advance is manufactured by Bruker which is not given at all. Shanghai is in China and this should be written. The provider of the sensor test system is not given. EC-CALS00 seems to be a type of the test instrument but not the provider. The cities of all these providers should be named.

Response 3: I apologize for our carelessness. The detailed information of chemicals and instruments is provided in the manuscript (line 102-105).

Point 4:  Compositional information: “A certain amount of the as-prepared H2SO4, SiO2 gel, and PVA solutions were mixed”:  This sentence is not appropriate. Even though the experimental findings will be applied, this should have been mentioned.

Response 4: Detailed preparation process of colloidal electrolyte was provided in the manuscript (line112-115).

Point 5: Page 3:

Experimental: It is not clear if the authors have manufactured a sensor or if they have used a sensor purchased from a Chinese company or if they simply used a standard three-electrode measurement system?  Is a sensor design regenerated from ref 20 or the company Zhengzhou Weisheng manufactures the bought sensor?

Response 5: We have completed catalyst preparation, electrolyte preparation, sensor assembly and performance testing. The sensor holders were purchased from Zhengzhou Weisheng Electronic Technology Co., Ltd., China. Electrode printing and post-processing were also produced by Zhengzhou Weisheng Electronic Technology Co., Ltd., China. Because electrode preparation consists of pretreatment of PTFE film, screen printing, ethanol washing, drying, water washing, drying and other processes, these processes are described in detail in our group's master thesis (Title: CO Electrochemical Gas Sensors Based on Water-based electrodes, (https://kns.cnki.net/KCMS/detail/detail.aspx?dbname=CMFD2012&filename=1012353871.nh). Our group is fully capable of manufacturing sensor electrodes with good performance. Because the preparation process is tedious, we commissioned the company to manufacture electrodes for us.

The sensor design is not from reference 20, but from the master's thesis of our group of students (reference 21). As mentioned above, reference 20 was misquoted in the manuscript.

In the revised manuscript,we provided the schematic diagram of the sensor structure with the photos of electrodes, sensor and gel (Figure 1, line134-136). The preparation process of electrode and the assembly process of sensor are briefly introduced (line124-130).

Point 6: Figures 1 and 2: TEM and SEM pictures are not revealing any detail. A higher magnification SEM picture is necessary for more detail and also the size of particles must be marked at TEM. No compositional or elemental hints are given. Pt black is of lose particles and can phase conductivity problems in real systems.

Response 6: New TEM and SEM photos were provided in Figure 3a, 3b. The HRTEM image and size distribution histogram were provided in Figure 3a and 3c (line 159-161). The crystal lattice was calculated to be 0.225 nm, which is assigned to a (111) plane according to the standard reference from Joint Committee on Powder Diffraction Standards (PDF01-087-0640) (line 164-168).

Point 7: Page 4:

Is it significant that Pt black has cubic structure? On the other hand, more important information is missing. No XRD data from fumed SiO2 or from the used gel SiO2. What types of Silica how influences the gel electrolyte properties.

Response 7: The crystal structure of Pt is one of the physical properties of Pt. However, the crystal structure cannot be related to the performance of the sensor. SiO2 is amorphous, so there is no XRD data of SiO2. Colloidal SiO2 and Fumed SiO2 are commonly used gelling agents (line 70-73). The colloidal formation mechanism is given in lines 194-200 of the manuscript (line 201-206).

Point 8: Ref 20 is on the Battery electrolytes.

Response 8: As mentioned above, reference 20 was misquoted in the manuscript. The reference [20] was corrected to be the master's thesis [21] (line 435-437).

Point 9: Figure 4: Effect of which silica? please indicate on the graph and also in the figure caption.

Response 9: Figure 4 shows the influence of fumed SiO2 content on gelling time (line 207-208).

Point 10: How are the reactions at WE, CE and total determined? It should be mentioned that it is assumed that the reactions occur and this knowledge is from the references 9 and 12. I appears that the authors themselves found these out.

Response 10: The electrode reactions are known by reference [9-11]. This was clearly stated in revised manuscript (line 236).

Point 11: Figure 5, 6, 7, 8, 10: The types of variants are consistently neglected in the figures as well as in the figure captions; SiO2 concentration, H2SO4, CO, sensor type and sensing test conditions (dry/wet/operating temperature, etc.), which two sensors, Pls indicate in details in graph and in the figure captions.

Response 11: The SiO2, H2SO4 concentrations and test conditions were added in Figure 5 (line 258-260) and Figure 6 (line 267-269). The test conditions and electrolyte composition in figure 7, 8 and 10 were added (line 279-281). Details of the variables in Figure 5, 6, 7, 8, 10 were given.

Point 12: Page 7:

It is not clear which electrochemical reaction occurs when CO injected, with what it reacts? and how this develops please indicate in the figure. There is only description of it.

Response 12: Many literatures have reported the electrode reaction of CO electrochemical sensor [3, 6, 9-11]. The CO electrode reaction is clear and reliable (line 236-239).

Point 13: Page 10:

There is no improvement against the commercial electrochemical CO-sensor. The manuscript fails to demonstrate the avoidance of long-term disadvantages of aqueous H2SO4 electrolytes that is provided with the applied Fumed SiO2 gel electrolyte.

Response 13: We did only compare the performance of the gel electrolyte and the H2SO4 electrolyte sensor under common environmental conditions. Sensor performance under extreme harsh environmental conditions is an important. We are conducting experiments on the impact of harsh environment on sensor performance.

Point 14: Conclusion: very weak.

Response 14: The conclusion part was rewritten. The advantages of gel electrolyte were emphasized, and the direction of further research was pointed out (line 362-373).

Point 15: May bring new advantages: what are these? And how and why it is provided? please indicate clearly.

Response 15: Unfortunately, we only compared the thermogravimetric curves of SiO2 gel solution and H2SO4 aqueous solution, and found that the water loss rate of SiO2 gel is significantly lower than that of aqueous H2SO4 solution (Figure 6, line 219-229).

Reviewer 3 Report

This manuscript addressed interesting subject. I recommend publication after some modifications.

  1. Title: Please include H2SO4 and PVA.
  2. Abstract: Please show linear range and detection limit.
  3. Indication of lethal level of CO concentration is better.

Author Response

Dear reviewer 3:

We would like to thank the thorough and constructive comments and suggestions on our manuscript. Those comments are all valuable and very helpful for revising and improving our paper, as well as the important to our researches. We are enclosing below all the detailed answers point by point. The revised parts are in red letters in our manuscript. Thank you very much for sharing your opinion and advice.

Point 1: Title: Please include H2SO4 and PVA.

Response 1: The title was changed to “SiO2-H2SO4-PVA gel electrolyte CO electrochemical gas sensor” (line 2).

Point 2: Abstract: Please show linear range and detection limit.

Response 2: The minimum detection limit was calculated as 2 ppm based on baseline data and sensitivity (line 21). The linear range is 2-500 ppm (line 21).

Point 3: Indication of lethal level of CO concentration is better.

Response 3: The death caused by carbon monoxide is not only related to the concentration of CO, but also related to the time of exposure to CO. 220 ppm CO may lead to carboxyhemoglobin reaching 30%, which is fatal [20]. However, we did not find data on exposure time leading to death in the literature. The dangers of CO were briefly described in the manuscript (line 76-82).

Unfortunately, there is no data of CO lethal level in the manuscript.

Reviewer 4 Report

General Comments:

The work presents a proposal of a SiO2 gel electrolyte to replace and overcome the usual liquid H2SO4 electrolyte for electrochemical measurement of CO concentration in air.

The manuscript is clear and well written, and the results show a very good performance of the gel electrolyte.

The authors should include an illustration of the electrochemical cell setup, showing the electrodes, the amperometric instrument, the gel, and how the CO gas enters the cell, close the electrode.

Specific Comments:

Lines 169-170: There is inconsistency between the text and Figure 5. The text mentions current values of uA, but the Figure shows values of nA.

Line 179: the text mentions “The sensor with 4.8% SiO2 had high sensitivity to CO”, but in line 176 and in Figure 5 one can see that the highest sensitivity occurs for 5.6% SiO2.

Line 238: The current density unit is  A.cm-2.

Line 286: Correct the last word: “conducted”.

Author Response

Dear reviewer 4:

We would like to thank the thorough and constructive comments and suggestions on our manuscript. Those comments are all valuable and very helpful for revising our paper, as well as the important to our researches. We are enclosing below all the detailed answers point by point. The revised parts are in red letters in our manuscript. Thank you very much for sharing your opinion and advice.

The authors should include an illustration of the electrochemical cell setup, showing the electrodes, the amperometric instrument, the gel, and how the CO gas enters the cell, close the electrode.

Response:

Figure 1 shows the schematic diagram of the sensor structure with the photos of sensor, gel and electrodes (line 134-136).

Figure 2 shows the schematic diagram of the sensor test system and the photos of gas mixing control unit, signal acquisition unit, computer control and storage unit of the sensor test system (line 150-154).

CO gas is diffused into the sensor cavity through capillary on the sensor,as shown in Figure 1 (line134). CO is oxidized to CO2 on the working electrode.

Point 1: Lines 169-170: There is inconsistency between the text and Figure 5. The text mentions current values of uA, but the Figure shows values of nA. 

Response 1: The current values in Figure 5 should be μA. The ordinate unit in Figure 5 was revised to μA (line 258).

Point 2: Line 179: the text mentions “The sensor with 4.8% SiO2 had high sensitivity to CO”, but in line 176 and in Figure 5 one can see that the highest sensitivity occurs for 5.6% SiO2.

Response 2: The final step in fabricating a sensor was to inject the electrolyte through the liquid injection hole at the bottom of the sensor, and then sealed the liquid injection hole to obtain an intact gas sensor. Because the diameter of hole is only 2 mm, and the viscosity of gelled electrolyte was very high. The liquid injection time was about 30 s. The electrolyte with high SiO2 content resulted in high viscosity of the electrolyte and long injection time. Therefore, we chose the colloidal electrolyte with lower viscosity (4.8% SiO2) rather than 5.6% SiO2 electrolyte.

The reasons for choosing 4.8% SiO2 electrolyte are given in the manuscript (line 253-257).

Point 3: Line 238: The current density unit is A.cm-2.

Response 3: Thank you very much. The current density unit is A.cm-2 (line 319).

Point 4: Line 286: Correct the last word: “conducted”.